# Infections in G6PD-Deficient Hospitalized Patients—Prevalence, Risk Factors, and Related Mortality

**DOI:** 10.3390/antibiotics11070934

**Published:** 2022-07-12

**Authors:** Diaa Alrahmany, Ahmed F. Omar, Salima R. S. Al-Maqbali, Gehan Harb, Islam M. Ghazi

**Affiliations:** 1Pharmaceutical Care Department, Directorate General of Medical Supplies, Ministry of Health, Muscat HCX8+9PP, Oman; diaa.alrahmany@yahoo.com; 2General Medicine Department, Suhar Hospital, Sohar 8MGH+JFH, Oman; farouqmail2000@yahoo.com; 3Department of Pathology and Blood Bank, Suhar Hospital, Sohar 8MGH+JFH, Oman; almoq96@yahoo.com; 4GH Statistics, Cairo 21524, Egypt; gehan.harb74@gmail.com; 5Arnold and Marie Schwartz College of Pharmacy, Long Island University, Brooklyn, NY 11201, USA

**Keywords:** G6PD deficiency, bacterial infections, prevalence, hospitalization, mortality, hospital-acquired infections, MDR-related infections

## Abstract

G6PD deficiency is a genetic disease that weakens the immune system and renders affected individuals susceptible to infections. In the Sultanate of Oman resides a high number of recorded G6PD cases due to widespread consanguineous marriage, which may reach 25% of the population. We studied the infection patterns and risk factors for mortality to provide antimicrobial stewardship recommendations for these patients. After obtaining ethical approval, a registry of recorded cases was consulted retrospectively to include G6PD-deficient adult patients admitted to Suhar hospital over 5 years with microbiologically confirmed infections. Patient demographics, health-related information, infection causes, treatment, and clinical outcomes were studied. Data were analyzed to describe infection patterns and risk factors. Several variables, including underlying comorbidities and hospitalization details, such as length of stay, admission to critical care unit, blood transfusion, or exposure to an invasive procedure, were statistically associated with the acquisition of multidrug-resistant and hospital-acquired infections. Meanwhile, these infections were associated with a high mortality rate (28%), significantly associated with the patient’s health status and earlier exposure to antimicrobial treatment due to previous bacterial infection. The high prevalence of G6PD deficiency among the Omani population should alert practitioners to take early action when dealing with such cases during infection that requires hospitalization. Strict infection control measures, Gram-negative empiric coverage, hospital discharge as early as possible, and potent targeted antimicrobial therapy in this patient population can ameliorate the treatment outcomes and should be emphasized by the antimicrobial stewardship team.

## 1. Introduction

Glucose-6-phosphate dehydrogenase (G6PD) is a crucial enzyme for the proper functioning of red blood cells (RBCs). Genes encoding G6PD enzyme production are found on the long distal arm of the X chromosome [1]. The G6PD enzyme stimulates the reduction of nicotinamide adenine dinucleotide phosphate (NADP) in the pentose phosphate pathway (PPP) to generate NADPH; the latter is a substrate for NADPH oxidase responsible for regenerating the antioxidant glutathione that protects red blood cells (RBCs) against oxidative stress [2].

Genetic abnormalities causing deficiency of the G6PD enzyme lead to uncontrolled premature hemolysis of RBCs triggered by viral or bacterial infections, sulpha-containing drugs, and certain types of food, manifested mainly as fatigue, pallidness, jaundice, shortness of breath, tachycardia, dark urine, and splenomegaly [3]. Clinical symptom severity in G6PD-deficient patients corresponds to the level of G6PD activity in the affected cells. In the absence of oxidative stress, even with substantially reduced enzyme activity, there may be few or no clinical symptoms.

A reduced NADPH pool causes reduced NADPH oxidase activity, leading to the defective production of reactive oxygen species (ROS)-related neutrophil extracellular traps (NETs) required for the antimicrobial activity of phagocytes and leukocytes, leading to recurrent bacterial and fungal infections [4,5,6]. As one of the forefront immune cells recruited during infections, neutrophils exhibit a wide range of mechanisms to counteract bacterial invasion via phagocytosis and the production of ROS, proteases, and NETs. As a result, any failure to recruit neutrophils to an infection site fosters the propagation of systemic infection [7,8].

Clinically, G6PD deficiency was found to be more prevalent in infected males than in the matched groups and suggested to be a predictor of hospitalization and severe infections [9].

A plethora of bacterial and fungal infections was identified in several studies that described the infections in G6PD-deficient patients, among which pneumonia, gastrointestinal, osteomyelitis, cerebrospinal, and septicemia were more common. Most of these infections were caused by *Chromobacterium violaceum* [10], *Staphylococcus aureus* [11], *Escherichia coli* [12], *Serratia marcescens* [13], *Acinetobacter baumannii* [14], *Klebsiella pneumoniae* [15], *Pseudomonas aeruginosa* [16], *Salmonella species*, *Staphylococcus epidermidis*, *Clostridium difficile* [17], and *Aspergillus species* [6].

G6PD deficiency is the most prevalent genetic enzyme deficiency affecting approximately 400 million individuals worldwide [18,19], with the highest prevalence in sub-Saharan Africa and the Middle East’s second-highest estimates [19]. Sultanate of Oman harbors one of the highest recorded cases of G6PD worldwide, almost 25% in males and 10% in females, due to the high rate of consanguinity marriage [20]. This opportunity may not exist in other communities due to the scarcity of G6PD deficiency cases to study and document the pattern of infectious diseases in patients suffering from this rare genetic blood disease. Landscaping the infection patterns and identifying patients at high risk of mortality may aid in developing clinical care algorithms that optimize treatment outcomes for this patient cohort.

## 2. Methods

### 2.1. Study Population

Genetically tested adult G6PD-deficient patients (>18 years) with infections confirmed by microbiological laboratory who were admitted over the period (1 January 2017, to 31 December 2021) to our tertiary care facility were included in this investigation. After the study was approved by the Ministry of Health’s Research and Ethical Review Committee, patient-relevant data was collected from the hospital’s electronic medical records.

We examined the patients’ age, gender, clinical symptoms of infection (to exclude patients with colonization), existing comorbid conditions, diabetes mellitus (DM), chronic renal failure (CRF), active malignancy, immuno-suppressed, chronic cardiac diseases (CCD), chronic respiratory disease (CRD), exposure to invasive procedures (endotracheal tube insertion, urinary catheterization, wound debridement, venous catheterization, lumbar puncture or similar procedures) during admission, 90-day prior exposure to surgery, and 90-day history of infections. Hospitalization details included diagnosis at admission, discharge status, length of stay (LOS), and admission ward. Microbiological details included laboratory-confirmed microbiological cultures, infection sites, specimen type, susceptibility pattern, resistance phenotype, prior infections, and concurrent infections. Only the first episode was selected for patients with several admissions with identical cultures. Patients with no Suhar hospital ID, patients with positive cultures who were not admitted, died before receiving a single dose of antibiotics, and pediatric patients (>18 years) were excluded.

### 2.2. Definitions

Hospital-acquired infections occurred ≥72 h of the admission date; all other episodes were considered community-acquired infections (CAI) [21]. Admission to an intensive care unit (ICU), cardiac care unit (CCU), or burn unit (BU) for more than 24 h is considered a critical care stay. A complete or partial resolution of infection signs, normalization of laboratory values of white blood cell count (WBC) and C-reactive protein (CRP), or negative culture of the exact source of the original infection was used to determine the clinical prognosis at the end of treatment. On days 14 and 28 of hospitalization, mortality was considered if the symptomatic patient had a positive culture and died before resolving infection signs.

The treatment of infection using a single antibiotic is considered monotherapy, while combined therapy is using 2 or more antibiotics with antimicrobial action against the causative organism during the infection episode.

Multidrug-resistant (MDR) infection was defined following CLSI 2010 M100-S20 guidance [22], as the isolate resistant to one antibiotic of three or more different antimicrobial classes. Carbapenem-resistant Enterobacterales (CREs) were phenotypically detected—according to CLSI—as the isolates that showed inhibition zones <23 mm with (ertapenem 10 μg or meropenem 10 μg) and tested resistant to one or more antibiotics in cephalosporin subclass III (e.g., cefotaxime, ceftazidime, and ceftriaxone). Confirmed CRE was reported as resistant for all penicillins, cephalosporins, carbapenems, and aztreonam.

### 2.3. Statistical Analysis

The data were analyzed using R software statistical programming language, version 3.6.2 (2019-12-12) (R Foundation for Statistical Computing platform). Median and interquartile ranges (IQRs) were used to describe numerical data and analyzed using linear regression analysis after the normality was tested using Shapiro–Wilk normality test. Categorical data are analyzed using binary logistic regression and expressed using *p*-values, odds ratios (ORs), and confidence intervals (CIs). All tests were two-sided; *p*-values < 0.05 are considered significant, at a 95% confidence level.

## 3. Results

Medical records of 3334 registered G6PD-deficient patients between 1 January 2017 and 31 December 2021 were reviewed; 2512 patients were excluded because they were <18 years when they had a laboratory-documented bacterial infection during the study period (2017–2021), while 620 other adult patients were excluded as they did not have any microbiological cultures or hospitalization details. The remaining 202 patients’ records were examined over the 5-year period, and (379) microbiological cultures corresponding to hospital admissions were recorded and studied. See Figure 1

### 3.1. Patients’ Demographics

The majority of the microbiological cultures belonged to male patients (69.9%). The study cohort’s median (IQR) age at admission was 59.9 (41–77), with patients’ ages evenly distributed around the age of 60 years (≤60 years, 50.1%, and >60 years, 49.9%). In total, 72% of the patients were diagnosed with an infectious disease upon admission to the hospital, and the vast majority of the patients were admitted to medical wards (38.5%) with a median (IQR) LOS 12 (5–31). The hospital stay ended in death in 27.7% of cases, with a high incidence of short-term deaths (14-day mortality of 52% of total mortality).

The vast majority of these patients (89%) were suffering from underlying chronic diseases, with a median (IQR) of 3 (2–4) chronic diseases; CCD (84%), DM (75%), and CRF (67%) were the most prevalent of these conditions. A total of 10.3% of the patients possessed a 90-day surgical history, 75.2% underwent invasive procedures during admission sessions, and 55.9% needed a blood transfusion of packed RBCs. Heparin/LMWH was prescribed for prophylaxis/treatment in 63.6% of the cases, inotropes in 32.5%, and vasodilators in 27.4%. Table 1 describes patient demographics.

### 3.2. Infection Patterns

The microbiological samples originated in almost an equal proportion from soft tissues (27%), urinary tract (25%), respiratory system (24%), and blood (23%). Gram-negative bacteria dominated the majority of infections (60%), with *Klebsiella* sp. (27%), *Pseudomonas* sp. (26%), *E. coli* (19%), *Acinetobacter* sp. (14%), and others (15%). Meanwhile, Gram-positive bacteria accounted for (28%) of the cases as follows: *Staphylococcus coagulase-negative* (CoNS) (37%), *Methicillin-sensitive Staphylococcus aureus* (MSSA) (21%), *Enterococcus* sp. (16%), *Methicillin-resistant Staphylococcus aureus* (MRSA) (11%), and *Streptococcus* sp. (14%). Fungal infections were detected in 8% of the samples, with *Candida albicans* accounting for 84% of the infections and other fungi accounting for the remainder. We also tracked severe acute respiratory syndrome coronavirus SARS-CoV-2 infections and found only 14 cases that required hospitalization. Polymicrobial infections occurred in 59% of the cases, among which concurrent infections with Gram-negative was 75%, with Gram-positive was 47%, with Fungi was 26%, and 3% with SARS-CoV-2.

Infections were caused by susceptible bacterial phenotypes in 54% of cases, with the remaining cases caused by resistant isolates, which were distributed as follows: MDR (20%), extended-spectrum β-lactamase bacteria ESBL (15%), CRE (7%), and MRSA (4%). While 56% of the infections were community-acquired (CAIs), hospital-acquired infections (HAIs) accounted for more than one-third of all cases, 44%. Within 90 days, a new infection with a different organism occurred in 36% of the cases. See Figure 2.

90-day infections prior to index admission occurred in 23% of cases, of which 55% were Gram-negative, 39% were Gram-positive, 23% were SARS-CoV-2, and 5% were fungal.

### 3.3. Susceptibility Pattern

*Acinetobacter* sp. (14%), *E. coli* (19%)*, Klebsiella* sp. (27%)*, P. mirabilis* (7%), and *Pseudomonas* sp. (26%) represented 93% of the total Gram-negative pathogens; they showed high susceptibility to colistin and tigecycline (98% and 92%, respectively), and moderate susceptibility to amikacin, meropenem, and piperacillin/tazobactam (~75%). Meanwhile, they showed higher resistance to cephalosporins, ciprofloxacin, and cotrimoxazole. See Figure 3.

### 3.4. Antimicrobial Treatment

A total of 42% had been exposed to antimicrobials 90 days prior to index admission, primarily cephalosporins (66%), β-lactams (44%), quinolones (41%), β-lactam/β-lactamase (37%), and other antimicrobials. A total of 66% of the study cohort received antimicrobial monotherapy, while 34% received combined therapy, mainly cephalosporin-based (41%), B-lactam/B-lactamase inhibitor-based (35%), piperacillin/tazobactam-based (26%), quinolones-based (12%), and vancomycin-based therapy (10%). Table 1 details the antimicrobial treatment received.

### 3.5. MDR-Related Infections

MDR-related infection was significantly associated to >28-day mortality [*p* < 0.026, OR: 2.44], prolonged LOS >14 days [*p* < 0.000, OR: 2.41], more than 2 comorbidities [*p* < 0.032, OR: 1.17], mainly CCD [*p* < 0.016, OR: 1.84], blood transfusion during admission [*p* < 0.001, OR: 2.01], infection with Gram-negative bacteria [*p* < 0.000, OR: 3.08], prior infection with SARS-CoV-2 [*p* < 0.009, OR: 3.69], concurrent infections with Gram-negative and fungal infections [*p* < 0.040, OR: 1.54] and [*p* < 0.000, OR: 2.96], respectively, and HAIs [*p* < 0.000, OR: 2.90]. MDR-related infections required combined therapy [*p* < 0.049, OR: 1.54], mainly piperacillin/tazobactam-based [*p* < 0.000, OR: 2.75], Meropenem-based [*p* < 0.000, OR: 6.16], Colistin-based [*p* < 0.000, OR: 8.35], and Tigecycline-based therapy [*p* < 0.022, OR: 6.18]. See Table 2.

### 3.6. Hospital-Acquired Infections (HAIs)

HAIs were significantly associated with bacteraemia [*p* < 0.033, OR: 0.58] and thus more related to 14-day mortality [*p* < 0.011, OR: 2.13], prolonged LOS > 14 days [*p* < 0.000, OR: 18.42], admission to critical care [*p* < 0.000, OR: 10.58], cumulative number of comorbidities [*p* < 0.010, OR: 1.21], especially CCD *p* < 0.003, OR: 2.11] and CRD [*p* < 0.001, OR: 2.38], infection with Gram-negative bacteria [*p* < 0.000, OR: 2.90], infection with MDR phenotypes [*p* < 0.000, OR: 4.10], and 90-day prior exposure to metronidazole [*p* < 0.046, OR: 2.21].

Monotherapies [*p* < 0.054, OR: 1.53] were more likely prescribed for HAIs. Mainly, the prescribed regimens were meropenem-based [*p* < 0.029, OR: 2.44], colistin-based [*p* < 0.000, OR: 7.07], and tigecycline-based therapy [*p* < 0.037, OR: 5.28]. Polymicrobial infections significantly occurred during HAI [*p* < 0.000, OR: 9.00], most likely Gram-negative [*p* < 0.000, OR: 7.60], Gram-positive [*p* < 0.000, OR: 2.97], and Fungal infections [*p* < 0.000, OR: 3.85]. See Table 2.

### 3.7. Fourteen-Day Mortality Risk Factors 

Fourteen-day mortality was significantly related to males [*p* < 0.000, OR: 6.55], patient age >60 years [*p* < 0.000, OR: 5.63], admission to critical care areas [*p* < 0.005, OR: 2.30], bacteraemia [*p* < 0.005, OR: 2.37], HAIs [*p* < 0.011, OR: 2.13], invasive procedures during admission [*p* < 0.001, OR: 10.51], cumulative number of comorbidities [*p* < 0.000, OR: 1.54], especially CCD [*p* < 0.010, OR: 3.19] and CRF [*p* < 0.004, OR: 2.69], and immunosuppressed patients or those with active malignancy ([*p* < 0.042, OR: 2.63] and [*p* < 0.005, OR: 4.84], respectively). See Table 3.

### 3.8. Twenty-Eight-Day Mortality Risk Factors

Twenty-eight-day mortality occurred more frequently in the following: patients >60 years [*p* < 0.002, OR: 6.97], prolonged LOS [*p* < 0.021, OR: 1.01], cumulative number of comorbidities [*p* < 0.003, OR: 1.72], especially CRD [*p* < 0.000, OR: 5.05], concurrent infection with either Gram-negative [*p* < 0.001, OR: 6.14] or Gram-positive bacteria [*p* < 0.000, OR: 6.26], 90-day history of exposure to antimicrobials [*p* < 0.037, OR: 2.59], especially cephalosporins [*p* < 0.020, OR: 2.80], β-lactams [*p* < 0.008, OR: 3.36], and macrolides [*p* < 0.004, OR: 4.41]. See Table 3.

## 4. Discussion

In our practice, identifying patients with this genetic blood disease that may affect their immune response to infections is an integral part of infection control plans and offering optimal therapeutic options pursuant to antibiotic stewardship efforts.

In the Arabian Peninsula, consanguineous marriage widespread within the same tribe resulted in a large spread of G6PD deficiency. Oman is one of the region’s countries with a high number of such cases [23]. Since our tertiary care hospital serves more than a third of the nation’s population and has a considerable number of registered cases of G6PD, we had an excellent opportunity to landscape infectious diseases in a decent number of hospitalized G6PD patients and identify the factors that may cause HAIs, infection with MDR bacteria, and the mortality rates associated with these infections.

Over five years, all adult G6PD-deficient patients admitted to the hospital with microbiological proof of infection at admission or during hospitalization were studied. The vast majority of microbiological cultures were isolated from male patients; a study by Rostami and colleagues corroborated this finding, perhaps due to linkage to the sex chromosome [24]. Gram-negative bacteria dominated the majority of infections (60%), mainly *Klebsiella* sp., *Pseudomonas* sp., *E. coli*, and *Acinetobacter* sp. Meanwhile, Gram-positive bacteria accounted for (28%) of the cases, mainly *CoNS*, *S. aureus*, *Enterococcus* sp., *MRSA*, and *Streptococcus* sp., while fungal infections were detected in 8% of the samples, with *C. albicans* accounting for 84% of the infections. Although we could not find any previous research that comprehensively identified bacterial infections in G6PD patients prior to writing this article, numerous reports for individual cases described the infectious pathogens observed in this study [6,11,12,13,14,15].

### 4.1. Risk Factors for MDR-Related Infections

Prolonged LOS contributed significantly to the acquisition of MDR-related infections; both have been linked together in several studies [25,26,27], leading us to believe that attempting early discharge, especially for this type of patient, may reduce the risk of acquiring MDR-related infections while also lowering the cost of hospitalization and treatment.

Patients admitted to critical care areas were almost 2-fold liable for acquiring MDR infection compared to other wards, which is consistent with the finding of Tosi and colleagues [28]. This may be explained by the extensive antibiotic pressure in these areas and other factors such as older age, chronic comorbidities, and suppressed immunity.

Patients with more than two comorbidities were more likely to acquire MDR-related infections; thus, the early medical stabilization of such cases may significantly impact clinical outcomes. Meanwhile, multiple studies statistically correlated the acquisition of MDR-related infections with blood transfusion during admission [29,30], which could be an opportunity to reduce the acquisition of MDR-related infections when adopting robust infection control measures, and commitment to clearly indicated rather than routine blood transfusion, especially since the hospital-acquisition of MDR bacteria in our cohort was very significant.

MDR Gram-negative bacteria were the primary causative organisms; this can be explained by a large proportion of patients having previously been exposed to Gram-negative infection, which may have resulted in the transfer of resistance determinants to current strains of bacteria. Patients with prior SARS-CoV-2 infection were more likely to acquire MDR-related infection; multiple retrospective studies duplicated the same finding [31,32,33]. A total of 65% of MDR infections were polymicrobial, mainly with Gram-negative and fungal infections.

### 4.2. Risk Factors for HAI

MDR pathogens were the leading cause of HAIs; patients with HAIs were suffering from chronic comorbidities and required admission to critical care, which explains the high rates of 14-day mortality related to HAIs. Increased antimicrobial treatment exposure increases the prevalence of virulent nosocomial bacterial phenotypes. Adopting better multidisciplinary infection control practices, regular disinfection of patient care equipment, reducing unnecessary admissions through updated outpatient practices, and increasing outpatient care resources contribute to a reduction in HAIs [34].

### 4.3. Risk Factors for Mortality

Age was consistently a risk factor for 14-day and 28-day mortality; patients >60 years had 6-fold higher mortality rates compared to those below 60 years; this is probably due to deteriorated health status and organ dysfunction, as well as immune deficiency caused by immune cell inefficiency in these patients due to G6PD.

The cumulative number of comorbidities significantly contributed to early and late-onset mortalities, mainly in patients with CCD, CRF, CRD, and immunocompromised patients (see Table 3 for details). Chang and colleagues identified illness severity, duration of mechanical ventilation, prior hospitalization, and underlying conditions as predictors of mortality [35], implying that concurrent comorbidities to infections must be clinically stabilized as soon as possible to improve outcomes.

Fourteen-day mortality was significantly related to bacteremia, primarily due to HAI; most of those patients needed to be admitted to critical care areas and required invasive procedures during admission; intuitively, patients admitted to critical care areas usually suffer a diminished health status, weakened immunity, and are more vulnerable to invasive maneuvers and highly virulent pathogens, which necessitate regular disinfection of the ICU environment and patient equipment, as well as optimal intubation practice and avoidance of unnecessary catheterization [34].

As for 28-day mortality, patients with polymicrobial infections were 6-fold liable to death compared to those with monomicrobial; isolates able to disseminate resistance determinants via horizontal gene transfer (HGT) may trigger the high acquisition of MDR infections, which eventually leads to death in immunocompromised patients, given that most of our study cohort suffered severe hemolytic anemia that required blood transfusion during hospitalization.

## 5. Conclusions

The high prevalence of G6PD deficiency among the Omani population should alert practitioners to consider the possible reduced immune status of this patient population. Evidence-based protocolization of hospital admissions, blood transfusion, and intubation is a crucial clinical step to reduce HAIs and MDR pathogen acquisition in this unique patient group.

Parallel to the early stabilization of underlying comorbid conditions, strict infections, control measures, Gram-negative empiric coverage, the early starting of potent targeted antimicrobial therapy, and hospital discharge at the earliest time possible, in general, and particularly in this patient population, can ameliorate the treatment outcomes and should be emphasized by the antimicrobial stewardship team.

## Figures and Tables

**Figure 1 antibiotics-11-00934-f001:**
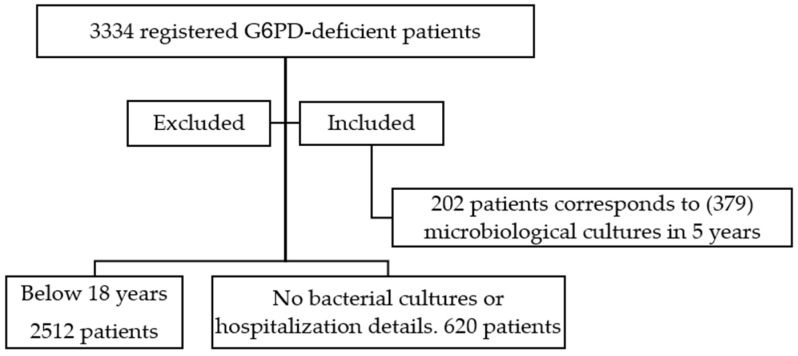
A chart of the patients screened for inclusion inn ther study.

**Figure 2 antibiotics-11-00934-f002:**
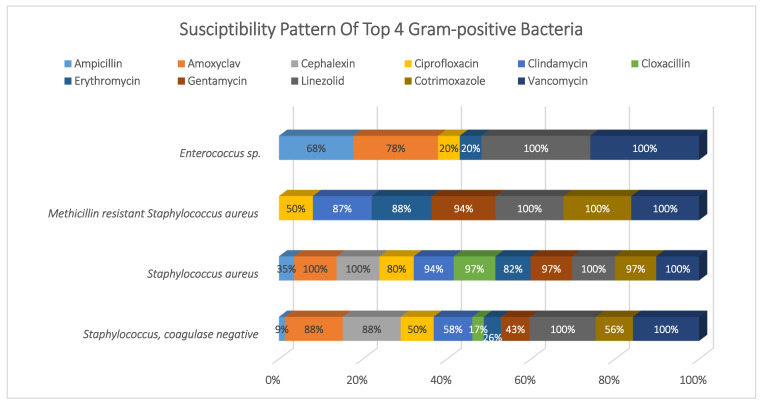
Susceptibility pattern of top 4 Gram-positive bacteria.

**Figure 3 antibiotics-11-00934-f003:**
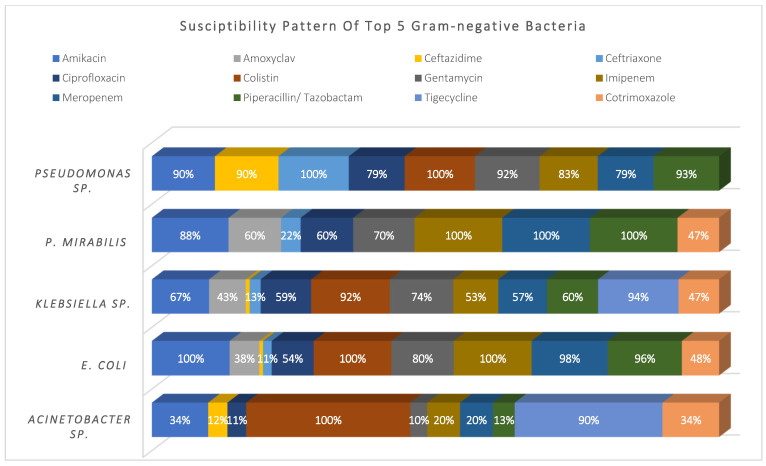
Susceptibility pattern of top 5 Gram-negative bacteria.

**Table 1 antibiotics-11-00934-t001:** Patient demographics and relevant clinical details.

	Included (*n* = 379)			Included (*n* = 379)	
	N	(%)		N	(%)
**Male**	265	69.9%	**Age on admission—Median (IQR)**	59.9	(41–77)
Female	114	30.1%	≤60 Years	190	50.1%
**Length of stay (LOS) Median (IQR)**	12	(5–31)	>60 years	189	49.9%
≤14 days	184	48.5%	**Admission Diagnosis**		
>14 days	195	51.5%	Infectious disease diagnosis	273	72.0%
**Discharge outcome**			Non-infectious diagnosis	106	28.0%
Death	105	27.7%	**Admission ward**		
Recovery	274	72.3%	Critical care area	117	30.9%
14-day mortality	55	52.4%	Medical ward	146	38.5%
28-day mortality	22	21.0%	Surgical ward	116	30.6%
>28-day mortality	28	26.7%	**Other Risk Factors**		
**Underlying Comorbid conditions**			Invasive procedure during admission	285	75.2%
Number of comorbid conditions	3	(2–4)	Need for blood transfusion	212	55.9%
Any comorbidity	338	89.2%	Surgery 90-day history	39	10.3%
Chronic Cardiac Diseases	282	83.4%	**Medication received**		
Diabetes	254	75.1%	Analgesics	283	74.7%
Chronic renal failure	228	67.5%	Proton pump inhibitor	269	71.0%
Others	179	53.0%	Heparin/LMWH	241	63.6%
Chronic Resp. Disease	70	20.7%	Diuretics	206	54.4%
Immuno-suppressed	24	7.1%	Cortico-steroids	124	32.7%
Sickle Cell	19	5.6%	Inotropes	123	32.5%
Active malignancy	14	4.1%	Vasodilators	104	27.4%
HIV follow-up	1	0.3%	Albumin	66	17.4%
**Culture sample type**			**Organism causing infections**		
Skin and soft tissue	103	27.2%	Gram-positive bacteria	107	28.2%
Urine	96	25.3%	Gram-negative bacteria	227	59.9%
Respiratory	91	24.0%	Fungal	31	8.2%
Blood	88	23.2%	SARS-CoV2	14	3.7%
Body fluids	1	0.3%	**Place of Acquisition**		
**Bacterial Resistant Phenotype**			Community-acquired infection	212	55.9%
Sens	181	54.2%	Hospital-acquired infection	167	44.1%
MDR	67	20.1%	90 days occurrence of any infection	136	35.9%
ESBL	50	15.0%	**90-day previous infections**	87	23.0%
CRE	24	7.2%	Gram-negative	48	55.2%
MRSA	12	3.6%	Gram-positive	34	39.1%
**Prior exposure to antimicrobials**	158	41.7%	SARS-CoV-2	20	23.0%
Cephalosporins	105	66.5%	Fungal	4	4.6%
β-lactams	70	44.3%	**Antimicrobial treatment**		
Quinolones	64	40.5%	Monotherapy	250	66.0%
β-lactam/β-lactamase	58	36.7%	Combined therapy	129	34.0%
Macrolides	34	21.5%	Cephalosporin-based	157	41.4%
Glycopeptides	33	20.9%	B-lactam/B-lactamase inhibitor-based	131	34.6%
Nitroimidazole	29	18.4%	Pip/Taz based	99	26.1%
Aminoglycosides	25	15.8%	Quinolones-based	45	11.9%
Tetracyclines	19	12.0%	Vancomycin-based	38	10.0%
Glycylcycline	3	1.9%	B-lactam-based	37	9.8%
Oxazolidinones	2	1.3%	Antifungal treatment	31	8.2%
Colistin	1	0.6%	Tetracycline-based	30	7.9%
**Concomitant infections**			Macrolide-based	28	7.4%
Polymicrobial infections (Yes)	224	59.1%	Meropenem based	28	7.4%
Gram-negative	169	75.4%	Colistin-based	24	6.3%
Gram-positive	106	47.3%	Aminoglycosides-based	22	5.8%
Fungal	59	26.3%	Tigecycline-based	10	2.6%
SARS-CoV-2	7	3.1%	Linezolid based	2	0.5%

**Table 2 antibiotics-11-00934-t002:** Variables related to hospital-acquired infections/acquisition of MDR infections (Binary logistic regression). Check Appendix A for comprehensive statistical values.

Included (*n* = 379)	N (%)	Hospital-Acquired Infections (*n* = 167)			MDR Infection (*n* = 153)		
		N	*p*	CI	N	*p*	CI
		(%)	OR		%	OR	
Age on admission—median (IQR)	59.9	59	0.399	(0.99, 1.01)	63	0.848	(0.99, 1.01)
	(41–77)	(36.5–77)	1.00		(41–77)	1.00	
Age > 60 years	189	81		(0.60, 1.36)	80		(0.78, 1.77)
	49.9%	48.5%	0.91		52.3%	1.18	
Length of stay (LOS) median (IQR)	12	34	0.000	(1.08, 1.13)	22	0.000	(1.01, 1.02)
	(5–31)	(17–60)	1.10		(8–41.5)	1.01	
LOS > 14 days	195	139	0.000	(10.92, 31.07)	94	0.000	(1.58, 3.67)
	51.5%	83.2%	18.42		61.4%	2.41	
Admission to critical care area	117	94	0.000	(6.23, 17.98)	64	0.000	(1.50, 3.66)
	30.9%	56.3%	10.58		41.8%	2.35	
Number of comorbid conditions Median (IQR)	3	3	0.010	(1.05, 1.40)	2	0.032	(1.01, 1.36)
	(2–4)	(2–4)	1.21		(0–3)	1.17	
Any comorbidity	338	149	0.982	(0.52, 1.94)	139	0.391	(0.68, 2.66)
	89.2%	89.2%	1.01		90.8%	1.35	
Chronic Cardiac Diseases	282	137	0.003	(1.29, 3.44)	124	0.016	(1.12, 3.02)
	83.4%	82.0%	2.11		81.0%	1.84	
Diabetes	254	104	0.082	(0.44, 1.05)	99	0.431	(0.54, 1.29)
	75.1%	62.3%	0.68		64.7%	0.84	
Chronic renal failure	228	108	0.112	(0.92, 2.13)	100	0.089	(0.95, 2.21)
	67.5%	64.7%	1.40		65.4%	1.44	
Chronic Resp. Disease	70	43	0.001	(1.39, 4.05)	34	0.123	(0.89, 2.54)
	20.7%	25.7%	2.38		22.2%	1.51	
Immuno-suppressed	24	15	0.066	(0.95, 5.22)	9	0.767	(0.37, 2.06)
	7.1%	9.0%	2.23		5.9%	0.88	
Active malignancy	14	9	0.131	(0.78, 7.17)	5	0.718	(0.27, 2.48)
	4.1%	5.4%	2.36		3.3%	0.81	
Invasive procedure during admission	285	137	0.000	(4.26, 15.59)	101	0.150	(0.88, 2.33)
	75.2%	82.0%	8.15		66.0%	1.43	
Need for blood transfusion	212	155	0.000	(5.14, 13.55)	121	0.001	(1.37, 3.08)
	55.9%	92.8%	8.34		79.1%	2.01	
Surgery 90-day history	39	15	0.458	(0.39, 1.53)	15	0.798	(0.46, 1.81)
	10.3%	9.0%	0.77		9.8%	0.91	
Infection due to Gram-positive bacteria	107	28	0.000	(0.21, 0.55)	38	0.195	(0.46, 1.17)
	28.2%	16.8%	0.34		24.8%	0.74	
Infection due to Gram-negative bacteria	227	123	0.000	(1.88, 4.49)	115	0.000	(1.96, 4.83)
	59.9%	73.7%	2.90		75.2%	3.08	
Community-acquired infections	212	*	*	*	62	0.35	(0.23, 0.53)
	55.9%	*	*		40.5%		
Hospital-acquired infections	167	*	*	*	91	0.000	(1.89, 4.43)
	44.1%	*	*		59.5%	2.90	
Gram-negative 90-day previous infection	48	16	0.328	(0.39, 1.37)	21	0.610	(0.64, 2.16)
	55.2%	18.4%	0.73		13.7%	1.17	
Gram-positive 90-day previous infection	34	10	0.474	(0.37, 1.58)	14	0.920	(0.51, 2.12)
	39.1%	11.5%	0.77		9.2%	1.04	
SARS-CoV-2 90-day previous infection	20	15	0.001	(3.59, 204.55)	14	0.009	(1.39, 9.84)
	23.0%	17.2%	27.09		9.2%	3.69	
Fungal 90-day previous infection	4	2	0.810	(0.18, 9.13)	2	0.695	(0.21, 10.65)
	4.6%	2.3%	1.27		1.3%	1.48	

CI: confidence intervals, OR: odds ratio IQR: Interquartile range, LOS: Length of stay, SARS-CoV-2: severe acute respiratory syndrome-coronavirus-2019. * Value can’t be produced by software.

**Table 3 antibiotics-11-00934-t003:** Risk factors for 14 and 28-day mortalities (Binary logistic regression). Check Appendix A for comprehensive statistical values.

Included (*n* = 379)	N (%)	14-Day Mortality (*n* = 55)			28-Day Mortality (*n* = 22)		
		N	*p*	CI	N	*p*	CI
		(%)	OR		(%)	OR	
Male	265	51	0.000	(2.31, 18.60)	16	0.768	(0.44, 3.04)
	69.9%	92.7%	6.55		73%	1.16	
Age on admission—Median (IQR)	59.9	74	0.000	(1.02, 1.06)	78	0.000	(1.02, 1.09)
	(41–77)	(65.81.5)	1.04		(69–84)	1.05	
>60 years	189	45	0.000	(2.74, 11.55)	19	0.002	(2.03, 23.95)
	49.9%	81.8%	5.63		86%	6.97	
Length of stay (LOS) Median (IQR)	12	15	0.043	(0.97, 0.99)	33	0.021	(1.00, 1.02)
	(5–31)	(4–22)	0.99		(26–39.3)	1.01	
Admission to the critical care area	117	26	0.005	(1.28, 4.11)	8	0.566	(0.53, 3.19)
	30.9%	47.3%	2.30		36%	1.30	
Admission to a medical ward	146	27	0.084	(0.93, 2.95)	14	0.017	(1.22, 7.29)
	38.5%	49.1%	1.66		64%	2.98	
Number of comorbid conditions Median (IQR)	3	3	0.000	(1.22, 1.95)	2	0.003	(1.19, 2.48)
	(2–4)	(2–4)	1.54		(2.8–4)	1.72	
Any comorbidity	338	54	0.000	(1.22, 1.95)	22	*	*
	89.2%	98.2%	1.54		100%	*	
Chronic Cardiac Diseases	282	49	0.010	(1.32, 7.7025)	20	0.086	(0.83, 15.81)
	83.4%	89.1%	3.19		91%	3.63	
Chronic renal failure	228	43	0.004	(1.37, 5.30)	17	0.100	(0.85, 6.52)
	67.5%	78.2%	2.69		77%	2.35	
Other comorbid conditions	179	35	0.009	(1.21, 3.95)	13	0.255	(0.69, 3.99)
	53.0%	63.6%	2.19		59%	1.66	
Chronic Respiratory Disease	70	12	0.490	(0.64, 2.58)	11	0.000	(2.09, 12.19)
	20.7%	21.8%	1.28		50%	5.05	
Immunosuppressed	24	7	0.042	(1.04, 6.68)	1	0.724	(0.09, 5.37)
	7.1%	12.7%	2.63		5%	0.69	
Active malignancy	14	6	0.005	(1.61, 14.54)	1	0.828	(0.16, 10.10)
	4.1%	10.9%	4.84		5%	1.26	
Invasive procedure during admission	285	53	0.001	(2.51, 44.01)	22	*	*
	75.2%	96.4%	10.51		100%	*	
Need for blood transfusion	212	36	0.126	(0.88, 2.89)	20	0.004	(1.98, 37.32)
	55.9%	65.5%	1.59		91%	8.59	
Bacteraemia	88	21	0.005	(1.29, 4.35)	5	0.955	(0.35, 2.71)
	23.2%	38.2%	2.37		23%	0.97	
Hospital-acquired infections	167	33	0.011	(1.19, 3.81)	14	0.063	(0.96, 5.70)
	44.1%	60.0%	2.13		64%	2.33	
Prior exposure to antimicrobials	158	20	0.387	(0.43, 1.39)	14	0.037	(1.06, 6.33)
	41.7%	36.4%	0.77		64%	2.59	
90-day exposure to Cephalosporins	105	16	0.804	(0.58, 2.04)	11	0.020	(1.17, 6.67)
	66.5%	80.0%	1.08		79%	2.80	
90-day exposure to β-lactams	70	5	0.060	(0.15, 1.04)	9	0.008	(1.37, 8.21)
	44.3%	25.0%	0.40		64%	3.36	
90-day exposure to β-lactam/β-lactamase	58	12	0.150	(0.83, 3.47)	8	0.007	(1.40, 8.79)
	36.7%	60.0%	1.69		57%	3.51	
90-day exposure to Macrolides	34	2	0.152	(0.08, 1.48)	6	0.004	(1.60, 12.15)
	21.5%	10.0%	0.34		43%	4.41	
Concomitant infections with Gram-negative	169	27	0.468	(0.69, 2.19)	18	0.001	(2.04, 18.51)
	75.4%	71.1%	1.24		82%	6.14	
Concomitant infections with Gram-positive	106	10	0.084	(0.26, 1.09)	15	0.000	(2.48, 15.85)
	47.3%	26.3%	0.53		68%	6.26	
Concomitant infections with Fungi	59	6	0.306	(0.26, 1.54)	10	0.000	(2.15, 12.78)
	26.3%	15.8%	0.63		45%	5.24	

CI: confidence interval, OR: odds ratio IQR: Interquartile range, LOS: length of stay. * Value can’t be produced by software.

## Data Availability

Raw data are available at Suhar Hospital database, upon written request and approval.

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
