# Peer review of "Infections in G6PD-Deficient Hospitalized Patients—Prevalence, Risk Factors, and Related Mortality"

_antibiotics, 2022, doi:10.3390/antibiotics11070934_

Round 1
Reviewer 1 Report
The paper needs an extensive English correction, better from a mother tongue.
Line 51: Better to change “generation” with “pool”
Pay attention to “G6PD” spelling, some sentences report G6DP, please fix it.
Lines 63-64: Rephrase the sentence because it is unclear.
Line 74: Change in “G6PD deficiency”
Line 81: please change with “Adults”
Line 86: please replace “;” with “:”
In the methods section: please fix the abbreviations. You should write the entire term followed by abbreviation in brackets. (Also, check the entire papers to change this)
Line 100: Please replace with “Hospital acquired infections”
There are too many typos, carefully check the entire paper (e.g. Enterobacterales, not Enterobacteralis)
Line 144: Please change ; with :
Line 159: Why did you write Staphylococcus term alone?
Line 164: please use the correct term SARS-CoV2 (and check the other parts in which it should be changed)
Lines 178-182: Please rewrite the sentence adding brackets since in that way the concept is unclear.
Please note that MRSA is a type of Staphylococcus, then you should not use in the same sentence (e.g. lines 262-263) both the terms. If you would like to differentiate, you could use MSSA and MRSA.
Line 275: Please change “Martina and colleagues” with “Tosi et al”
Lines 281-284: Please rephrase the sentence since it is totally unclear
Line 285: Please rephrase the sentence using plural terms
It would have been useful to better characterize the relationship between G6PD deficiency and bacterial infections, in terms of immune system function and systemic response.
Discussion section should be more explicative, especially as regards the correlation between results and G6PD deficiency.
In addition, G6PD deficiency and SARS-CoV2 infection should be further analyze; it may be useful to read and add these papers: 10.3892/wasj.2020.64; 10.2174/1568026622666220516111122; 10.1038/s41598-021-98712-3
Author Response
|
§ The paper needs an extensive English correction, better from a mother tongue. |
|
Thank you for your advice; a mother-tongue English speaker has thoroughly revised the article. |
|
§ Line 51: Better to change “generation” with “pool” |
|
Generation changed to “pool” |
|
§ Pay attention to “G6PD” spelling; some sentences report G6DP; please fix it. |
|
The term G6PD is reviewed all over the manuscript |
|
§ Lines 63-64: Rephrase the sentence because it is unclear. |
|
The sentence is rephrased to describe the most common infections and causative pathogens reported in the literature. “A plethora of bacterial and fungal infections was identified in several studies that described the infections in G6PD-deficient patients, among which pneumonia, gastrointestinal, osteomyelitis, cerebrospinal, and septicemia were more common. Most of these infections were caused by Chromobacterium violaceum [10], Staphylococcus aureus [11], Escherichia coli [12], Serratia marcescens [13], Acinetobacter baumannii [14], Klebsiella pneumoniae [15], Pseudomonas aeruginosa [16], Salmonella species, Staphylococcus epidermidis, Clostridium difficile [17] and Aspergillus species [18].” |
|
§ Line 74: Change in “G6PD deficiency” |
|
Changed to “Scarcity of G6PD deficiency cases to study and document the pattern of infectious diseases.” |
|
§ Line 81: please change with “Adults” |
|
Changed to “Genetically tested adult G6PD deficiency patients (>18 years) with infections confirmed by the microbiological laboratory.” |
|
|
|
|
|
§ In the methods section: please fix the abbreviations. You should write the entire term followed by the abbreviation in brackets. (Also, check the entire paper to change this) |
|
Abbreviations all over the manuscript are put between brackets when first time spelled out. |
|
§ Line 100: Please replace with “Hospital acquired infections” |
|
Changed to ïƒ “Hospital-acquired infections occurred ≥ 72 h of the admission date; all other episodes were considered community-acquired infections (CAI)” |
|
§ There are too many typos, carefully check the entire paper (e.g. Enterobacterales, not Enterobacteralis) |
|
Thank you for your advice; a mother-tongue English speaker has thoroughly revised the article. |
|
§ Line 144: Please change ; with : |
|
Changed to ïƒ “The vast majority of these patients: 89%, were suffering from underlying chronic diseases” |
|
§ Line 159: Why did you write Staphylococcus term alone? |
|
Changed to ïƒ “of the cases as follows: Staphylococcus coagulase-negative CoNS (37%)” |
|
§ Line 164: please use the correct term SARS-CoV2 (and check the other parts in which it should be changed) |
|
SARS-CoV-19 changed to SARS-CoV2 all over the manuscript |
|
§ Lines 178-182: Please rewrite the sentence adding brackets since in that way the concept is unclear. |
|
Thank you for the advice: sentence rephrased “Acinetobacter sp., E. coli, Klebsiella sp., P. mirabilis, and Pseudomonas sp. represented (86%) of the total Gram-negative pathogens; they showed high susceptibility to colistin, and tigecycline (98%) and (92%), respectively, moderate susceptibility to amikacin, meropenem, and piperacillin/tazobactam (~75%), meanwhile they showed higher resistance to cephalosporins, ciprofloxacin, and cotrimoxazole.” |
|
§ Please note that MRSA is a type of Staphylococcus, then you should not use in the same sentence (e.g. lines 262-263) both the terms. If you would like to differentiate, you could use MSSA and MRSA. |
|
Thank you for the comment; we really wanted to differentiate between MSSA and MRSA. The sentence is rephrased as per the comment. |
|
§ Line 275: Please change “Martina and colleagues” with “Tosi et al” |
|
Changed to “consistent with the finding of Tosi and colleagues.” |
|
§ Lines 281-284: Please rephrase the sentence since it is totally unclear |
|
Sentence rephrased Meanwhile, multiple studies statistically correlated the acquisition of MDR-related infections with blood transfusion during admission [30, 31], which could be an opportunity to reduce the acquisition of MDR-related infections when adopting robust infection control measures, and commitment to clearly indicated rather than routine blood transfusion, especially since the hospital-acquisition of MDR bacteria in our cohort was very significant. |
|
§ Line 285: Please rephrase the sentence using plural terms |
|
Sentence rephrased MDR Gram-negative bacteria were the primary causative organisms; this can be explained by a large proportion of patients having previously been exposed to Gram-negative infection, which may have resulted in the transfer of resistance determinants to current strains of bacteria. |
|
§ It would have been useful to better characterize the relationship between G6PD deficiency and bacterial infections, in terms of immune system function and systemic response. |
|
Studying the connection between illness and susceptibility to infection is undoubtedly a good job, and that is what we are currently trying to do as we attempt to connect the laboratory results of blood cell analyses, indicators of inflammation, and organs’ functions to the severity of cases, and mortality rates. When we’re done drafting it, it would be great if you could evaluate it so we can get your input on this. We briefly touched on the connection between G6PD deficiency and bacterial infections in terms of cell-mediated immune system functions because the article primarily focuses on landscaping the pattern of infection and risks for mortality in G6PD-deficient patients rather than the clinical pathology of the disease.
|
|
§ Discussion section should be more explicative, especially as regards the correlation between results and G6PD deficiency. |
|
Discussion has been modified as per all reviewer comments collectively |
|
§ In addition, G6PD deficiency and SARS-CoV2 infection should be further analyze; it may be useful to read and add these papers: 10.3892/wasj.2020.64; 10.2174/1568026622666220516111122; 10.1038/s41598-021-98712-3 |
|
Thank you for enriching our knowledge with these articles relating the severity of SARS-CoV2 infections to G6PD deficiency. We already cited one of them, and we will use the rest in an ongoing work describing the infection’s treatment outcomes in this special patient group. |
Reviewer 2 Report
1) The full name of MDR should be provided when it first appears in the Abstract section. And the full name of LOS should also be provided when it first appears in the main text.
2) The current Abstract section is a bit verbose. I would suggest the authors to reorganize the Abstract section to make it more concise.
3) There are total 9 keywords in the current manuscript. I would suggest the authors to reduce the number of keywords.
4) The patients included in this work were 202, and they corresponded to 379 microbiological cultures. The legend of Table 1 suggested that this table should provide detailed information about patients (i.e., 202 patients). However, the clinical number shown in Table 1 (patient demographic and relevant clinical details) were 379. For example, the authors stated that 72% (i.e., 273 shown in Table 1) of the patients were diagnosed with an infectious disease upon admission to the hospital. Such number (i.e., 273) was larger than the number of total patients included in this work. How to explain such result?
5) The authors stated that Polymicrobial infections occurred in 59% of the cases, among which concurrent infections with Gram-negative was 75%, with Gram-positive was 47%, with Fungi was 26%, and 3% with SARS-CoV-19. The percentage sum of concurrent infections with Gram-negative (i.e., 75%), Gram-positive (i.e., 47%), Fungi (i.e., 26%), and SARS-CoV-19 (i.e., 3%) was 151%, which exceeds 100%. How to explain?
6) The authors stated that Gram-negative bacteria dominated the majority of infections (60%), with Klebsiella Sp. (27%), Pseudomonas sp. (26%), E. coli (19%), Acinetobacter sp. (14%), and others (15%). However, the percentage sum of 27%, 26%, 19%, 14% and 15% was 101%, which exceeds 100%. How to explain?
Moreover, the authors stated that Acinetobacter sp., E. coli, Klebsiella sp., P. mirabilis, and Pseudomonas sp. represented 86% of the total Gram-negative pathogens. However, the sum of the percentage of Klebsiella Sp. (27%), Pseudomonas sp. (26%), E. coli (19%), and Acinetobacter sp. (14%) has reached 86%. The percentage of P. mirabilis was 0%? How to explain.
7) The authors should carefully check all data in this manuscript.
8) The authors should provide clearer conclusions.
Author Response
|
§ The full name of MDR should be provided when it first appears in the Abstract section. And the full name of LOS should also be provided when it first appears in the main text. |
|
MDR: spelled out in the abstract Line 93: length of stay (LOS) first appears and is spelled in the main text. |
|
§ The current Abstract section is a bit verbose. I would suggest the authors reorganize the Abstract section to make it more concise. |
|
As per the journal instructions for authors, the abstract should be about 200 words maximum. The abstract should be a single paragraph and follow a structured abstract style but without headings, which is fulfilled in our abstract. However, we have further shortened the abstract. |
|
§ There are total 9 keywords in the current manuscript. I would suggest the authors to reduce the number of keywords. |
|
Thank you for your advice; the number of keywords is reduced. |
|
§ The patients included in this work were 202, and they corresponded to 379 microbiological cultures. The legend of Table 1 suggested that this table should provide detailed information about patients (i.e., 202 patients). However, the clinical number shown in Table 1 (patient demographic and relevant clinical details) were 379. For example, the authors stated that 72% (i.e., 273 shown in Table 1) of the patients were diagnosed with an infectious disease upon admission to the hospital. Such number (i.e., 273) was larger than the number of total patients included in this work. How to explain such result? |
|
Thank you Since diverse sample sources, different organisms, varied susceptibility patterns, and separate hospitalization episodes were required for inclusion, our analysis included many cultures that might relate to the same patient.
As per your illustration, the number of patients admitted with an infectious disease diagnosis pertains to the hospitalization period or to positive microbiological cultures, not to the entire number of patients included in the study. (For example, in 273 out of 379 cultures, the patient was admitted due to a diagnosis of an infectious disease.) |
|
§ The authors stated that Polymicrobial infections occurred in 59% of the cases, among which concurrent infections with Gram-negative was 75%, with Gram-positive was 47%, with Fungi was 26%, and 3% with SARS-CoV-19. The percentage sum of concurrent infections with Gram-negative (i.e., 75%), Gram-positive (i.e., 47%), Fungi (i.e., 26%), and SARS-CoV-19 (i.e., 3%) was 151%, which exceeds 100%. How to explain? |
|
As many patients have been exposed to concurrent polymicrobial infections with many species of bacteria or fungi, adding the percentages of instances so that the sum is equal to 100% cannot be applied in this situation. |
|
§ The authors stated that Gram-negative bacteria dominated the majority of infections (60%), with Klebsiella Sp. (27%), Pseudomonas sp. (26%), E. coli (19%), Acinetobacter sp. (14%), and others (15%). However, the percentage sum of 27%, 26%, 19%, 14% and 15% was 101%, which exceeds 100%. How to explain? |
|
In this instance, the ratios were rounded mathematically to make them more suitable for inclusion in the text and more comprehensible for the reader, which led the percentage to exceed 100%. |
|
§ Moreover, the authors stated that Acinetobacter sp., E. coli, Klebsiella sp., P. mirabilis, and Pseudomonas sp. represented 86% of the total Gram-negative pathogens. However, the sum of the percentage of Klebsiella Sp. (27%), Pseudomonas sp. (26%), E. coli (19%), and Acinetobacter sp. (14%) has reached 86%. The percentage of P. mirabilis was 0%? How to explain. |
|
We appreciate your accurate remark regarding the overall % in this situation, which has been changed as follows in the text. Acinetobacter sp. (14%), E. coli (19%), Klebsiella sp. (27%), P. mirabilis (7), and Pseudomonas sp (26%) represented (93%) of the total Gram-negative pathogens. |
|
§ The authors should carefully check all data in this manuscript. |
|
Thank you for the remark; the authors have carefully reviewed the manuscript’s ratios, numbers, and comparable data. |
|
§ The authors should provide clearer conclusions. The conclusion has been modified. |
Reviewer 3 Report
Review on: „Infections in G6PD Deficient Hospitalized Patients; Prevalence, Risk Factors, and Related Mortality“ by Alrahmany et al.
General comments: This manuscript deals with infections of patients with glucose-6-phosphate dehydrogenase deficiency in Omar. The topic is interesting and only few findings are available so far. The study design is well thought out from a scientific point of view and the data analysis is well implemented. Furthermore, the manuscript is written in an understandable way and therefore easy read. I therefore recommend the publication of the manuscript in the present form.
Author Response
|
General comments: This manuscript deals with infections of patients with glucose-6-phosphate dehydrogenase deficiency in Omar. The topic is interesting and only few findings are available so far. The study design is well thought out from a scientific point of view and the data analysis is well implemented. Furthermore, the manuscript is written in an understandable way and therefore easy read. I therefore recommend the publication of the manuscript in the present form. |
|
Your feedback was wonderful since it inspired the team working on this manuscript to implement the improvements the other reviewers suggested as quickly as feasible. We sincerely appreciate you for the positive emotions that we had after reading your upbeat comment. |
Round 2
Reviewer 1 Report
The authors improved the manuscript and fixed the typos/mistakes.
Reviewer 2 Report
The authors have answered all of my comments, and I would suggest acceptance for this manuscript.